# How Do Consumers’ Food Values across Countries Lead to Changes in the Strategy of Food Supply-Chain Management?

**DOI:** 10.3390/foods10071523

**Published:** 2021-07-01

**Authors:** Jisung Jo, Eon-kyung Lee

**Affiliations:** Korea Maritime Institute, Port Research Division, Busan 49111, Korea; jisungjo@kmi.re.kr

**Keywords:** consumer-centric, food supply chain, food value, consumer preferences, big data

## Abstract

Although one of the main goals of supply-chain management is to maximize consumer values, the research to date has mainly focused on the supply side. In the case of the food industry, understanding consumer needs and maximizing its utility are essential. In this study, we analyze consumers’ 12 meta-values (e.g., safety, taste, health, price, environment, etc.), then suggest the strategy of food cold-chain management satisfying consumers’ perception. We focused on consumers from three countries in Asia: Korea, China, and Japan. The survey was conducted with over 1000 consumers in those three countries, and a random parameter logit model was utilized to determine the importance of each food value that could affect consumers’ food choice. Similarities and differences were both found in share of preference of each food value across countries. While safety is one of the top three values in all three countries, naturalness and nutritional value ranked among the top three only in China. To propose the consumer-centric strategy of food cold-chain management, we investigated the relationship between each food value and each node of supply chain based on the big data analysis. It shows that consumers prefer when the entire supply chain is managed where each node is organically connected with each other instead of individual nodes being managed separately. Further, strategies for food cold-chain management should be developed differently by country, incorporating differences of consumers’ preferences on food value. These results would motivate governments and companies related to food cold chain to reconsider their marketing strategies on the import and export food market.

## 1. Introduction

Transitioning from the third industrial revolution into the current era of the fourth industrial revolution has altered the circumstances around supply-chain logistics. Access to technology, like the Internet, 3D printing, big data, artificial intelligence, and IoT, among others, has given consumers more opportunities than ever before to reveal their preferences on products. From the supply perspective, knowing consumers’ preferences helps suppliers to produce diversified or customized small-quantity production. In this manner, the focus of today’s supply-chain management is moving from a product-oriented era to a consumer-oriented era. Today’s companies have become aware of the importance of consumers’ satisfaction and strive to enhance the value of the end-consumers [1]. Selen and Soliman [2] mentioned that the rapid spread of technology—specifically, the Internet—has caused a shift in power away from the supplier towards the consumer. Christopher and Ryals [3] and Canever, Trijp, and Beers [4] showed that new technology and consumers’ different preferences motivated the emergence of a demand-focused chain.

Fisher [5] emphasized that the role of market mediation is as important as that of the supply chain’s physical function, including production, transporting, and inventory storage. The purpose of market mediation proposed by Fisher [5] is matching what consumers want to buy to the variety of products at the marketplace. Vollmann and Cordon [6] mentioned that demand chain management—starting with the end-customer to the supplier—is the most promising avenue to improve the competitiveness of a company. Vollmann et al. [7] stressed the changes from supply to demand focus in the supply-chain management as one of four major issues that need to be highlighted for business executives. Christopher and Ryals [3] insisted that to achieve a sustainable world, the supply chain needs to be designed from the customer backward instead of the factory outward, making it responsive to customer demands and reducing waste and returns.

Previous literature has insisted on the importance of identifying consumers’ value and preferences on market products to improve the effectiveness of the supply chain. There have been attempts to interpret demand-centric, agri-food supply chains with this concept [3,4,8]. In this research, we would like to investigate consumers’ food values in three different countries—South Korea, China, and Japan—that might show different preferences and estimate the relationship between food value and supply chain node to propose consumer-centric food supply-chain management strategy.

### 1.1. Background on Food Value

According to Rokeach [9], a value is defined as “an enduring belief that a specific mode of conduct or end-state of existence is personally or socially preferable to an opposite or converse mode of conduct or end-state of existence.” In his research, he introduces the instrumental value, which relates a preferable mode of behavior that achieves a particular end state and terminal value, which relates to preferable end states of existence. Using these values, he referred to a value system as “an enduring organization of beliefs concerning preferable modes of conduct or end-states of existence along a continuum of relative importance.” From Rokeach’s value research, a means-end chain theory [10,11] has been developed. Gutman [11] introduced a means-end chain theory, which links perceived product attributes (A) to consequences of the product (C) to personal values (V). It offers a mechanism for understanding what values are important to consumers to motivate their purchasing decisions based on the attributes. This model is meaningful in how it investigates perceptions of consumers and finds a linkage to their decisions. The means-end chain theory has been widely used in the field of marketing and psychology to identify factors affecting individuals’ decision-making process [12]. This model could be applied to any field of industry, and there are some studies applying the model to explain the linkage between consumers’ food consumption of any segment in the supply chain and their value.

There are many previous papers showing the linkage between consumers’ values and their consumption. Costa, Dekker, and Jongen [13] reviewed literature in the usage of the means-end chain theory in the food area and showed the model can improve the action ability of consumer-oriented food product design and the level of coordination between R&D and marketing. Sorenson and Henchion [14] conducted in-depth, one-to-one laddering interviews based on the means-end chain theory and found the relationships between consumers’ cognitions with regard to high-pressure-processed, chilled, ready meals. Bitzios, Fraser, and Haddock-Frasers [15] utilized a means-end chain model to reveal key attributes to be included in the food choice experiment. Lin, Fu, and Chen [16] constructed consumers’ cognitive structure toward services provided by classy restaurants based on the integration of means-end chain and balance theories. The means-end chain theory shows the linkage between the deeper motivations between women and men and three potential food hazards: mycotoxins, pesticides, and irradiation [17].

Lusk and Briggeman [18] utilized the means-end chain theory to define a set of consumers’ enduring beliefs that are likely to be relatively stable over time. They identified 11 meta food values that stably drive consumers’ preference for food product attributes; the 11 food values represent personal value (V), which links to the consumers’ food choice about the attributes of food products (A). According to Lusk and Briggeman [18], the list of food values could be generated slightly differently than the original to adjust for different situations. In this study, we would like to investigate consumers’ food values across three different countries to identify how personal values affect the food supply chain. Table 1 represents the definition of 12 food values; the list is identically applied across all the countries.

### 1.2. Background on Big Data Analysis

Big data analysis allows atypical data, such as those from social media, the Internet, journals, and media news, to be analyzed; hence, it is widely used in analyzing consumer attributes, measuring demand for policy, and developing corporate marketing materials. In this regard, studies on supply-chain management (SCM) started to recognize the importance of utilizing the potential role of social media and applying big data analysis to establish the consumer-centric marketing strategies. Chae [19] proposed a new framework of analysis (Twitter analytics) to develop additional insights on the potential role of Twitter for supply-chain practices (e.g., demand shaping, new product development, supply-chain risk management) and research. The implication is extracted from a research direction based on 22,399 tweets and meta data collected with the hash tag #supplychain from Twitter and uses three methodologies: descriptive analytics (DA), content analytics (CA) integrating text mining and sentiment analysis, and network analytics (NA).

Choi [20] assessed quick response programs for fashion with social media observation, demand forecast updating, and rational retailers. According to the result from the social media data analysis, positive product review influences the value of quick response. In addition, its impact is mediated by the fashion retailer’s prior attitude towards the market demand.

Govindan [21] analyzed big data analysis cases in logistics and supply-chain management based on research papers for a period of six years and three months (2012–March 2018), as revealed in Scopus. The study analyzed the articles of technology-driven tracking strategies, financial performance, and implementation issues and supply-chain capability maturity with big-data-driven supply chains.

Application of big data analysis in the cold-chain research field is relatively early in stage. Big data analysis is used to find measures to optimize production while maintaining safety and quality standard of the entire food supply chain. A great amount of research is done in Agri-BIGDATA where big data is applied to develop a smart agriculture system. For instance, intelligent communication or AR technology is used to deliver, in real time, accurate guidelines, like information on nitrogen related to propagation or cultivation of agricultural products to farmers based on big data so that producers can be assisted in efficient decision making [22]. Kiran S, et al. [23] put forth the algorithm model to match producers and consumers of perishables in the supply chain using electronic exchange and global matching programs with big data analysis. The suggested matching model is more realistic and transparent to the users compared to many e-supply chain portals, as it uses a mathematical algorithm that ensures the allocation with high profits in the entire cold chain.

Jin et al. [24] analyzed keywords related to food safety in 686 research papers in 4 literature databases, including Electronics Engineers (IEEE), Science Direct, Scopus, and Google Scholar, dating from 2015 to 2020. It showed that big data analysis is a successful application that projects, monitors, and controls food safety in cold chain through four steps of content analysis, econometric analysis, recommendation system, and machine learning. Singh et al. [25] used social media data from Twitter to identify issues in the existing supply chain and logistics management in food supply. This study uses hierarchical clustering with multiscale bootstrap resampling and text analysis with support vector machine (SVM) for big data analysis. Moreover, this paper analyzes positive and negative preferences of consumers on taste, color, smell, food safety, etc., of beef and applied this to resolving issues in beef supply-chain management.

Although application of big data analysis is still in the early stage on food safety issues and consumers’ preference in food supply chains, it can be an effective measure to provide insight for projection, such as customer behavior analysis, trend analysis, and demand prediction in various steps of food supply chains and to assist participants in the supply chain with real-time decision making. Against this backdrop, in this study, we would like to suggest a food cold-chain management strategy for each country covered in this paper by identifying the relationship between each food value and each node of food supply chain based on big data analysis.

## 2. Materials and Methods

Figure 1 shows the flowchart for all the steps of this study. The first step is about investigating the relative preference of food value by countries. For this, we conducted an online survey and utilized a random parameter logit model, details of which are described in Section 2.1 and Section 2.2. For the second step, we calculated the relationship between each core food value and each supply chain node. Lastly, based on these results, we derived the consumer-centric food supply-chain management strategy.

### 2.1. Survey

We conducted an online survey and obtained 1103 completed responses in panels maintained by Gallup in three different countries; 379 (34.4%) in South Korea, 363 (32.9%) in China, and 361 (32.7%) in Japan. Gallup maintains panels, a group of consumers who have agreed to participate in online surveys in these three countries, and we asked for the sample that could represent democratic characteristics of each country. Additionally, over 70% of respondents in Korea and China are in either the “Primary shopper” category or “Family members buy food at the same rate” category. Around 50% of consumers in Japan also responded that they are the main shopper in their household. This implies that our sample could represent general grocery shoppers in each country. To improve participants’ understanding of the survey, it was conducted in each country’s respective language; professional translators were involved in the translation process. Table 2 indicates the summarized characteristics of respondents. Across countries, over 80% of the participants responded that they are the primary shopper for their household or equally shopped with their family, which shows the representativeness of the sample for this study. A total of 50.2% of participants are female, and 63.5% have a college degree. Most respondents are between 18 and 64 years old.

Followed by Lusk and Briggeman [18], we utilized the best-worst scaling to determine the relative importance of 12 food values. A balanced, incomplete block design (BIBD) was utilized to assign 12 values to the choice sets. Each food value appears the same number of times (four) with every other value. The survey consists of the eight choice sets with six alternatives. Participants were required to choose the best and the worst food value. Figure 2 shows an example of eight best-worst questions used in the survey.

### 2.2. Econometrics Analysis—Random Parameter Logit Model

Since we would like to see how the importance each individual places on each food value affects their preferences for food, a random parameter logit model (RPL) was utilized. To analyze the best-worst scaling, we followed the approach of Lusk and Briggeman [18]. They conceptualized that consumers choose two items that maximize the differences between the most important value and the least important value. In this case, participants have 30 (6 × 5) possible pairs to choose in a question. The chosen pair is considered the best-worst combination, which maximizes the importance differences out of 30 possible alternatives. The latent importance of value *j* for individual *i* is
(1)$$\gamma_{ij}=\bar{\gamma_{j}}+\sigma_{j}\mu_{ij},$$
where $\bar{\gamma_{j}}$ is the mean of $\gamma_{j}$, which is the location of valued *j* on the underlying scale of importance; $\sigma_{j}$ is the standard deviation of $\gamma_{j}$ in the population; and $\mu_{ij}$ is the random term normally distributed with mean zero and unit standard deviation for *j* value. This specification implies that the importance of food values *j* is assumed to be normally distributed with mean $\bar{\gamma_{j}}$ and standard deviation $\sigma_{j}$.

The probability that *i*th consumer chooses item *j* and *k* as the best and worst, respectively, is
(2)$$\mathrm{Prob}\;\left(\;j\;\mathrm{is}\;\mathrm{chosen}\;\mathrm{best}\;\mathrm{and}\;k\;\mathrm{chosen}\;\mathrm{worst}\right)=\frac{exp^{\gamma_{ij}-\gamma_{ik}}}{\sum_{l=1}^{6}\sum_{m=1}^{6}exp^{\gamma_{il}-\gamma_{im}}-6}$$

This specification assumes that the probability that the difference in $I_{ij}$ and $I_{ik}$ is greater than the other 29 possible differences in the choice set. To estimate random parameters, we utilized standard Halton sequences draws. Train [26] found that the simulation variance in the estimated parameters with Halton sequences is considerably smaller than random draw.

Based on the RPL estimate, we can calculate share of preference of each food value. Share of preference for value j represents the forecasted probability that j food value is chosen as most important.
(3)$$\mathrm{Share}\;\mathrm{of}\;\mathrm{preference}\;\mathrm{for}\;\mathrm{value}\;j=\frac{exp^{\hat{\gamma_{j}}}}{\sum_{l=1}^{J}exp^{\hat{\gamma_{l}}}}$$

It provides an intuitive interpretation. If the share of preference for j food value is twice as large as that of *k* food value, it can be said that j value is twice as important as *k* value.

## 3. Results

### 3.1. Food Values

The importance of each food value is estimated by RPL and relative to novelty. Table 3 shows the estimated parameters by each country. On average, safety is the most important food value to Korean consumers and significantly more important than novelty. Taste is the second most important food value, and nutritional value, origin, and price come next. Relative to other countries, Korean consumers consider appearance as the least important food value when they purchase, but it is still statistically more important than novelty. Social values, such as animal welfare, fairness, and environmental impact, tend to be considered less important food values in Korea.

China shows a similar pattern with Korean consumers. Safety is the most important food value, and nutritional value, naturalness, and taste come next. On average, people consider animal welfare, origin, appearance, and fairness as relatively less important food values. Interestingly, origin is a relatively less important value to Chinese consumers, whereas it is located near the top rankings in both Korea (the fourth most important value) and Japan (the fourth most important value).

The most important food value in Japan is safety, which is the same in South Korea and China. Taste, price, and origin are the next important food values, and, on average, significantly more important than novelty. The relatively less important food values are environmental impact, animal welfare, and fairness. Across countries, social values are not considered as relatively more important than other values. The food values directly related to food quality and welfare, such as safety, taste, and nutritional value, are more important to consumers in all three countries. Intriguingly, price is of only intermediary importance in South Korea and China, while it is the third most important value in Japan. One way of interpreting this is that the price of food products is more influential in Japan than in South Korea and China.

Unfortunately, we cannot directly interpret the estimates of the RPL model. Thus, this study calculates the share of preferences of each food value, which indicates the forecasted probability that each food item is chosen the most. Table 4 reports that 28.1% of consumers in South Korea would choose safety as the most important food value. Taste has the next highest share of preferences, with 21.5% of people considering taste as the most important food value when they purchase foods. A total of 12.6% of consumers in Korea believe origin to be the most important value, which is about half as important as food safety. While more than 60% of people would choose safety, taste, and origin as the most important food value, people who would pick the remaining food values, such as environmental impact, animal welfare, fairness, appearance, or novelty, account for less than 3%.

In China, the top three food values that people chose as the most important are safety, naturalness, and nutritional value. Interestingly, the share of preferences of safety is 44.4%, which is almost three times as important as naturalness and nutritional value. Also, it is almost two times larger compared to Korea (28.1%) and Japan (21.0%). Furthermore, we can recognize that 16.5% of people in China would rate naturalness as the most important value, which is more than three times and five times larger than Korea and Japan, respectively. This demonstrates how important safety and naturalness food values are in China. Less than 2% of consumers would choose most of the remaining food values: environmental impact, convenience, appearance, origin, fairness, and novelty. Interestingly, while origin is the third most important food value in Korea, it would be picked by only 1.4% of people as the most important value in China.

Japan also shows different trends than South Korea and China. Price has the largest share of preferences, with 23.1% of Japanese on average believing price to be the most important value. This is more than three times the figures in Korea and China, which are 7.3% and 4.2%, respectively. The second and third most important values to people in Japan are taste and safety. Similar to the results of South Korea and China, less than 3% of people would rate appearance, environmental impact, fairness, animal welfare, and novelty as the most important food value in Japan.

In this research, we investigate the relationship between six food values, including safety, taste, origin, naturalness, nutritional value, and price, and each node in the supply chain. This is because these six food values are predominantly considered the most important food values in the three countries. Over 80% of respondents in Korea, China, and Japan tend to choose these food values as the most important values.

### 3.2. Big Data Analysis

Typical big data analysis is conducted in the following order: first, selecting the topic and related keyword(s); second, collecting data; and third, conducting analysis. Extracting data from atypical data, such as those from news, blogs, and journals, requires text mining methodology with which data is processed to fit specific purposes.

To investigate the relationship between each food value and each node of supply chain, we utilized big data analysis. Data used in the analysis are from blogs; news of 341 media companies, including CNN and BBC, over the last 6 years (from 2015 to 2020); and 499 articles from Emerald Insight and Science Direct (Elsevier) over the last 10 years (from 2010 to 2020). The data was collected with Saltlux’ TORANDO. It is a multipurpose big data collection program with which users can collect massive amounts of the data they want from various dynamic deep webs, including blogs and news, in automatic and parallel manners in real time. TORANDO is one of the most powerful big data collection engines that can collect massive data from the web as well as news, RSS, Twitter, Facebook, and other social media.

In this study, we conducted qualitative analysis by estimating the frequency of key words that could be interpreted as demonstrating the level of consumers’ interests. Table 5 summarizes the frequency of keywords—“farm”, “process&food”, “package&food”, “transportation&food”, “storage&food”, “retail&food”, and “customer&food”—that represent main food cold chain nodes. It was found that, of the 7 nodes of the food cold chain (farm, process, package, transportation, storage, retail, and customer), consumers’ interest was highest in farm (40.5%) followed by process (20.3%) and customer (20.0%). In the case of journals that represent academic interest, process (25.2%), farm (15.9%), storage (15.7%), customer (15.6%), and transportation (14.4%) nodes have been studied more frequently than package (6.7%) and retail (6.6%). General consumers (see data source from news and blog) tend more to consider farm (78.6%), process (42.8%), and customer (39.2%) nodes compared to package (14.3%), transportation (10.4%), storage (8.8%), and retail (5.8%). Interestingly, the trend of academic and general consumers’ interest on food cold chain nodes are similar. Thus, in this research, we try not to separate any data set for analysis.

We calculated the frequency of combined keywords of food values and supply chain nodes. Table 6 shows the frequency of each node appearing with the six food values, and Table 7 summarizes the frequency of each food value shown with each node. Across the food values, process and customer, which are the end recipient of food, showed the largest share. The food value price appeared 257,182 times, and of that, customer node accounted for 32.9%. This can be interpreted that the node where price is most mentioned is customer.

The node that has direct impact on safety was process, and food safety issues occur mainly when food is contaminated during processing and distribution. For example, in March 2017, Food Yellow 4, whose use had been banned (as it was known to cause heart diseases, asthma, and ADHD-like behavior in children) in black tea sold by China’s food company, Three Squirrels (San Zhi Song Shu, 三只松鼠), was detected, and its sales were banned. The nodes that are sensitive to food value taste are: customer, which finally chooses food at the grocery store, and process, which may change the taste of the products. The node that was found to be sensitive to origin was the process node, where the origin of food can change, and the customer node, where people decide to buy by looking at the origin. The process node was also sensitive to nutritional value and naturalness.

The food value that had highest interest across all supply chain nodes was price followed by safety. In particular, 66.1% of frequencies of the retail node was with price, which is the highest share compared to that of other nodes. This might be because final food price is decided at the retail level.

Table 8 shows Pearson correlation coefficients between cold chain nodes. We found a strong, positive relationship between nodes, which indicates that each node is organically connected with each other. Particularly, stronger relationships among customer, process and transportation nodes have been shown. For the farm node, it has strong correlation with process node (0.92), while having weaker correlation with package and retail nodes. Process was strongly correlated with every node of supply chain bar retail, and it can be inferred that when consumers’ interest in process nodes increase (or decrease), storage (0.97), transportation (0.94), and customer (0.93) nodes also increase (or decrease). Customer node has strong correlations, having 0.81 or over coefficients with all the supply chain nodes, and it is deemed that it is more strongly correlated with type and condition of process (0.93), package condition (0.91), and the level of control of temperature and humidity during transportation (0.96). The type and condition of food package (package) seems to affect the type of process, mode of transportation, and purchasing trend of customer node. Transportation is highly related to process, package, and customers, all of whose coefficients were over 0.9. Food storage had strongest correlation with process (0.97), followed by farm (0.87), transportation (0.89) and customer (0.88).

We estimated Pearson correlation coefficient of the six food values (Table 9): price, safety, taste, nutritional value, origin, and naturalness. The result shows that correlation among food values all surpassed 0.79, which is higher than that among food supply chain nodes, and in particular, price and taste had a very strong relationship with the other food values. Price, which was mentioned the most in all the nodes of food supply chain, has a very strong positive relationship with all the food values, with all the coefficients exceeding 0.89. Price was most strongly correlated with taste (0.96), followed by nutritional value and naturalness (0.94), origin (0.93) and safety (0.89). Food safety, which was second only to price, was most strongly correlated with origin (0.95).

Under the assumption that achieving each food value would necessitate a different cold-chain management strategy, we calculated the Pearson correlation coefficient between six food values and seven supply chain nodes, the result of which is summarized in Table 10. Overall, the Pearson correlation coefficients between the six food values and each node of supply chain were over 0.7, showing a strong, positive relationship. This means that the higher the interest in food value that affects food choices, the stronger the interest in each node and vice versa. This may indicate that, in order to satisfy the consumers’ requirement for food values that they recognize, participants in the food cold chain need to develop the cold-chain management strategy for the entire supply chain instead of each node.

The coefficients of the price, which is the most frequently mentioned food value in our data set, are all over 0.7 against all the nodes. The correlation coefficients of price in food value between each node of process, farm, storage, customer, and transportation are 0.98, 0.95, 0.93, 0.93, and 0.92 respectively. From this result, we could find that the factors carrying the heaviest impact in food price are cost of production and logistics. In order to sell food at an appropriate price satisfactory to end-purchasers, it is necessary to optimize the cost of production and logistics through technology innovation in the supply chain.

The nodes that were talked about the most in food safety include customer (0.97) and transportation (0.96), and all of its coefficients were over 0.7, implying that safety impacts the entire supply chain as much as price. Correlation analysis revealed that consumers felt transportation (0.96), package (0.94), and process (0.92) are closely related to food safety. From this, we could identify that the most important factors in food safety are keeping proper temperature and humidity during transportation, maintaining freshness with packaging, and preventing contamination from harmful substances during processing. Such cases of safety being compromised during transportation or processing are easily seen in media news. For example, in Korea in October 2020, influenza vaccination had to be suspended after it was found that the proper temperature was not maintained during the transportation of vaccines. There are cases of food products being distributed after being contaminated during processing. In September 2020, some sterilized weaning food sold in Korea was found to contain bacteria exceeding the allowable limit and an alien, hair-like substance.

The nodes that are strongly correlated with taste are farm (0.94) and process (0.94), as the farm node is related to raw produces, and the process node enhances taste of food. The analysis results show that taste of food may differ following the level of management during storage (0.91) and transportation (0.85). Naturalness and nutritional value are strongly correlated with farm and process. It is deemed that, to preserve the naturalness and nutritional value of the food products, it is important to make sure no alien substance is added in the processing process, and no nutritional value is destructed during storage.

The nodes that have strong positive relationship with origin turned out to be process (0.95), transportation (0.96), storage (0.92), and package (0.91). Imported agricultural products are relatively more affordable compared to local ones. As such, it happens occasionally that imported agricultural products are transformed into local products at processing factories in the course of import (transportation), processing, warehouse (storage), and labeling (package). For instance, in Korea, some cheap imported products (fern, bellflower root, sesame, beef, pork, etc.) are disguised as locally produced. Preventing this will require more systemized management from the import stage to labelling stage.

### 3.3. Supply-Chain Management Strategy

The common food value that consumers in Korea, China, and Japan consider most important and are more sensitive to is safety, and as it has strong positive relationship with all the nodes, enhancing safety will require management of the entire supply chain. In particular, as the share of preference of safety in China is 44.4%, which is about the double the figure in Korea or Japan, items that are related to food safety in the supply chain should be given more attention. Traceability has become an important issue of the global food supply chain with increased food safety concerns and the globalization of food production and distribution [27]. Food contamination related to food safety can occur at various points throughout the supply chain and can be successfully prevented by identifying the source of contamination through the traceability of the food chain [28]. As shown in Table 11, it is reasonable to maintain and trace appropriate temperature and humidity pursuant to the guidelines for each product from purchasing the raw produces of perishables to processing, packaging, storage, and transporting them to customers in order to maintain the safety of food.

It is widespread in the industry that companies in food supply chains are suffering from high losses due to food deterioration. Food deterioration increases both economic and environmental costs in supply chains. As reported by FAO [29], one-third of food produced for human consumption is lost or wasted globally, which amounts to about 1.3 billion tons per year. One of the applicable options to reduce food deterioration is to invest in preservation technologies during process, storage, transportation, and retail. Another option could be vertical cooperation, where the entire supply chain acts like a company to reduce production and transportation lead time and optimize inventory and sale strategies. For example, a leading supermarket in eastern China, Suguo, operates several large-scale distribution centers by integrating with downstream sellers to reduce inventory [30]. A total of 23.1% of Japanese consumers put food price at the top of the list, which is about three times higher than the figure of Korean (7.3%) and Chinese (4.2%) consumers. It can be assumed that the overall cost of the supply chain in Japan is high due to higher cost of logistics and labor compared to other countries, making consumers there react to price more sensitively. To resolve this, Japan will need to take measures to reduce raw produce price by innovating production technology, lower product cost by adopting processing methods to minimize defect rate and bring down logistics cost with integrated operation of food warehouse and shared transportation-delivery system, as shown in Table 12.

Around the globe, natural products are in high demand, as they are more concerned about health and wellness and think organic foods have beneficial ingredients for health [31]. Food naturalness is a key trend in recent studies of the food industry and is influenced by different supply chain factors. Naturalness related to farming practices is organic production without pesticides, hormones, and antibiotics considering animal welfare, and customers with a preference for naturalness look for organic food products and ingredients. Naturalist customers who are willing to pay a higher organic premium perceive the chemical processes in relation to a food ingredient to be bad. Additionally, customers who prefer food naturalness are sensitive to GM (genetically modified) content of food and its labeling in storages [32]. In China, after the 2008 Chinese milk scandal (三鹿奶粉污染事件), frequent incidents were related to the addition of harmful substances and fake food, such as fake rice, fake egg, and flour with added pulverized lime, making Chinese consumers anxious about processed food. Therefore, unlike in Korea or Japan, consumers in China appreciate naturalness more. To get the supply of food made with clean, raw produces without food additives in China, it is necessary to intensively trace and manage the processes where alien substances might be added, including farm, process and storage nodes (Table 13). We can expect that managing and linking producers, processors, and warehouses verified with blockchain may enhance consumers’ trust towards these food products.

With the improvement of the PPP (GDP per capita) in Korea and Japan compared to China, consumers began to consider taste more important, ranking taste as the second most important value. To provide the tasty food that consumers are satisfied with, it is necessary to find fresh, raw produce, process it to preserve the original taste and maintain freshness, store it in warehouses to keep it fresh, and deliver it promptly when customers want, as shown in Table 14. For example, according to gourmet coffee specialists, the unique, intrinsic value and taste of single-origin coffee is obtained by managing the processing of harvested coffee beans into quality coffee, including all the steps in process: roasting, grinding, and brewing. Premium coffees are maintained through the gustatory certification of the unique and irreplaceable flavor profiles of single-origin coffees by cupping experts, cupping standards, and rigorous procedures employed by the Cup of Excellence, which has become a key mechanism for locating and certifying single-origin coffee [33].

PricewaterhouseCoopers (PwC) points out that half of beef sold in China under the Australian label is not Australian beef. As awareness of food fraud increases, more Chinese consumers buying Australian meat products are searching for authentic products. However, they often cannot identity the country of origin and the quality of the product. Overcoming this requires shared responsibilities amongst agricultural and supply chain actors and the use of tracking and tracing technologies, such as blockchain [34]. In particular, consumers in Korea and Japan tend to regard it as important to manage the origin of products imported from China and other countries where prices are relatively lower. Maintaining the origin of import requires confirming where the raw produces were produced (farm), checking the origin label on the package (package), and selecting the mode of transportation considering the product attributes and logistics cost (transportation), as shown in Table 15. It is also important to trace the products so that imported products are not turned into locally produced ones in the course of processing and storage (Table 15).

Many studies have indicated that healthy (nutritional) food and a healthy diet can effectively prevent obesity and chronic diseases [35]. Interestingly, unlike in Korea (11.0%) or Japan (4.3%), nutritional value is one of the top three food values in China (13.8%), as shown in Table 4. One of the reasons might be because the 2008 Chinese milk scandal (三鹿奶粉污染事件) has left a damaging effect on the perception and purchasing behavior of Chinese consumers. In order to survive the fierce competition among dairy processors, companies involved in the scandal made the mistake of adding melamine to raw milk to increase its protein content [36]. Protein is not only the source of essential amino acids, but in some cases, it also plays an important role in an individual’s health and well-being. However, single-plant protein has limited nutritional values. Studies showed that mixtures of two or more plant protein sources are critical to obtain highly nutritious foods. The mixture strategy to get high nutritional value is highly dependent on processing conditions, ratio, and concentration of the blends and their interactions with the other components in the process node as well as diversity of sources and purity of the ingredients in the farm node [37]. To produce reliable, nutritious, and healthy food, natural plants and animals should be managed to be fresh, nutritive, and functional supplements that provide humans with basic health requirements in the farm node (Table 16). In the process node (Table 16), it is necessary to manage compounds, such as melamine, which can potentially have a fatal effect on humans, and enhance the relationship between farms and processors interested in producing healthy food in the industry-value chain. Additionally, temperature, humidity, and atmosphere should be managed properly in the warehouse to prevent destruction of nutrients, as shown in Table 16.

## 4. Discussion

As the development of fourth industrial revolution technology, such as artificial intelligence (AI) and big data, enabled in-depth analysis of consumer preference explaining their food choice, establishing consumer-centric, cold-chain management strategies became the core competitiveness factor of companies [1,2,3,4,5,6,7]. Firstly, in this research, we estimated the importance of meta food value, which links to consumers’ food choices in three different countries. Although South Korea, China, and Japan share some similarity in the culture, the share of preference of each food value differs by country. This means that the firms engaged in food cold chain need to establish different food cold-chain management strategies to incorporate consumers’ needs in the respective countries. Macready et al. [38] conducted an online survey in five European countries, including Germany, France, Poland, Spain, and the UK, to develop a conceptual model and investigated the relationship between consumers’ trust in the food chain actors and their confidence. One of their main findings was that there are differences in levels of both trust and confidence in the integrity of food products and technology across countries, and these differences in confidence still exist under the situation they control for differences in trust. That is, it is true that trust explains consumers’ confidence; however, other explanatory factors, such as cultural differences and geo-political histories, are also needed. Yang et al. [39] investigated consumers’ food values on imported fruits and vegetables in Japan, Taiwan, and Indonesia. They showed that there are both similarities and differences in the importance of food value among countries. Across the countries, safety is the most important food value, while the importance of some food values, such as origin and price, differs by country. Although research-target countries and food products are different from those of this research, the results are consistent with this study. Safety is the universal concern in this study, while other food values are considered differently according to countries.

Secondly, we calculated the correlation between six food values (price, safety, taste, nutritional value, origin, naturalness) taken from the top three values from the three countries and each node of the food supply chain. Interestingly, the six core food values have a strong relationship with all nodes of the cold chain. It implies that, in order to satisfy consumers’ preference on the six food values, it is necessary to manage the entire cold chain, not each node separately in the cold chain. This is also witnessed in the relevant studies. Bishara [40] stated that supplying safe medicine to patients requires temperature and humidity control and monitoring as well as quality management from the perspective of the entire supply chain, not the individual node. M. Goransson et al. [41] mentioned that waste reduction in the food industry and efficient and safe management of cold chain requires continuous monitoring and controlling of temperature in the entire food cold supply chains (FSCs), ranging from production to retail. That is, it can be reasoned that the strategy to manage the entire supply chain instead of each individual node is necessary for efficient pricing, food safety, and taste improvement in the food supply chain.

Lastly, we suggest a consumer-centric food cold-chain management strategy for each food value and country. Across countries, maintaining safety value within the food cold chain is found to be most essential; thus, many companies and governments have proposed devices, platforms, and policies to ensure the traceability of food products. In practice, Walmart China launched, in 2019, a blockchain traceability platform to build up consumers’ trust in safety, quality, and authenticity of their food products [42]. Korea and Japan have implemented a livestock traceability system to improve the quality of livestock and prevent food safety crisis [43,44,45]. Further, in 2019, the Korean government conducted a pilot project constructing a blockchain-based beef traceability system to improve trustworthiness of information and reduce trace time [46]. This strategy could be applied to ensuring the authenticity-related (origin and naturalness) food values as well. Traceability system and blockchain technology-based food cold-chain system, such as IBM Food Trust, could guarantee not only pursuing freshness (safety) of food but also uncovering food fraud along the supply chain [47].

It is an old debate of which one comes first: taste or health (nutritional value) [48,49,50]. Interestingly, 16.5% of consumers in China tend to choose healthiness (nutrition) as the most important food value, which is almost four to five times larger compared to the figure of Korea (4.9%) or Japan (3.1%). However, as for the taste value, around 21% of people in Korea and Japan would choose taste as the most important value, while the number is only 8.7% in China. This means that global companies will have to adopt different management strategies to enter these three different markets. For the Chinese market, process, package, and storage strategy need to be established to emphasize a healthy image of food products, while in the markets in Korea and Japan, the strategy to make products look tasty is more important. Since a company must establish a strategy to maximize profits under budget constraints, it is essential to identify the characteristics of each market and find a way to efficiently meet consumer needs.

These results are meaningful in that we found a relatively stable explanation of the relationship between consumers’ perception and cold-chain management and the management strategies to approach Asia’s food market. Now that global companies engaged in food cold chain know why consumers in Asian countries have different preferences on food products, they need different marketing strategies by country to dominate the local food market. Also, this research showed the first attempt to link consumers’ perception to food cold-chain management strategy using big data analysis. However, since we measured the frequencies and Pearson correlation coefficients of our data, the interpretation is somewhat limited. To have a more fruitful interpretation, conducting sentiment analysis, which might classify the data into positive, negative, and neutral class, could be a good approach. This will make it possible to interpret not only the interest of a certain key word but also consumers’ sentiment of it.

## 5. Conclusions

In this research, we investigated consumer food value, which influences food choice, and determined the relationship between each core food value in Asian countries and each node of the supply chain. Based on these results, we proposed the consumer-centric food supply-chain management strategy by food value and country. This study is expected to become a cornerstone of consumer-centric supply-chain management research. For the future research, it would be meaningful to conduct research which establishes the consumer-centric strategy by individual food products (e.g., beef, pork, milk, egg, apple) or characteristics (e.g., GM food, organic food). Further, it would be interesting to see how consumers’ trust in the food supply chain affects each food value and real food choices.

## Figures and Tables

**Figure 1 foods-10-01523-f001:**
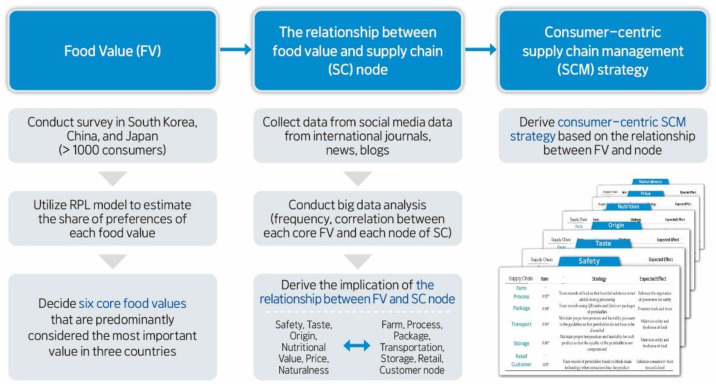
Flowchart of this study.

**Figure 2 foods-10-01523-f002:**
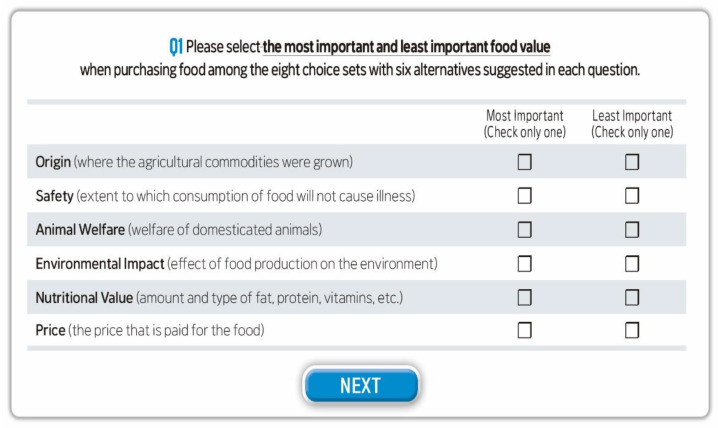
Example of best-worst question.

**Table 1 foods-10-01523-t001:** Food values and descriptions.

Food Value	Descriptions
Origin	if the agricultural commodities were grown locally or in a foreign country
Safety	the consumption of the product does not cause illness
Animal welfare	welfare of the animals in animal husbandry
Environmental impact	effect of food production on the environment
Nutritional value	amount and type of fat, protein, vitamins
Price	the price you pay for the food
Fairness	if farmers, manufacturing companies, retail, and consumers benefit equally
Novelty	the food you buy is something new that you did not taste before
Convenience	how easily and quickly the food is ready to eat
Naturalness	produced without modern technologies and ingredients
Taste	the taste of the food in your mouth
Appearance	if the food looks appealing

**Table 2 foods-10-01523-t002:** Socio-demographic characteristics of the sample.

Characteristics	Category	Korea	China	Japan	Total
Total		379	363	361	1103
		(34.4)	(32.9)	(32.7)	(100)
Primary Shopper	Primary shopper	265	284	197	746
		(35.5)	(38.1)	(26.4)	(100)
	Not the primary shopper	54	30	125	209
		(25.8)	(14.4)	(59.8)	(100)
	Family members buy food at the same rate	60	49	39	148
		(40.5)	(33.1)	(26.4)	(100)
Gender	Male	186	184	179	549
		(33.9)	(33.5)	(32.6)	(100)
	Female	193	179	182	554
		(34.8)	(32.3)	(32.9)	(100)
Age	18–24 years old	71	71	70	212
		(33.5)	(33.5)	(33.0)	(100)
	25–34 years old	78	70	69	217
		(35.9)	(32.3)	(31.8)	(100)
	35–44 years old	85	79	80	244
		(34.8)	(32.4)	(32.8)	(100)
	45–54 years old	73	73	70	216
		(33.8)	(33.8)	(32.4)	(100)
	55–64 years old	57	52	50	159
		(35.8)	(32.7)	(31.4)	(100)
	65–74 years old	12	18	22	52
		(23.1)	(34.6)	(42.3)	(100)
	Over 75 years old	3	0	0	3
		(100.0)	(0.0)	(0.0)	(100)
Marital Status	Single	176	88	181	445
		(39.6)	(19.8)	(40.7)	(100)
	Married	179	262	157	598
		(29.9)	(43.8)	(26.3)	(100)
	Divorced/Widowed	24	13	23	60
		(40.0)	(21.7)	(38.3)	(100)
Education	Less than High School	10	9	14	33
		(30.3)	(27.3)	(42.4)	(100)
	High School	118	45	142	305
		(38.7)	(14.8)	(46.6)	(100)
	2-Year College Degree	51	63	56	170
		(30.0)	(37.1)	(32.9)	(100)
	4-Year College Degree	174	225	132	531
		(32.8)	(42.4)	(24.9)	(100)
	Master’s Degree/ Professional Degree	26	21	17	64
		(40.6)	(32.8)	(26.6)	(100)
Income	Under 18,107	36	116	119	271
		(13.3)	(42.8)	(43.9)	(100)
	18,117~36,224	130	143	95	368
		(35.3)	(38.9)	(25.8)	(100)
	36,234~54,341	97	44	63	204
		(47.5)	(21.6)	(30.9)	(100)
	54,351~72,458	50	30	35	115
		(43.5)	(26.1)	(30.4)	(100)
	72,468~90,575	39	20	16	75
		(52.0)	(26.7)	(21.3)	(100)
	Over 90,585	27	10	7	44
		(61.4)	(22.7)	(15.9)	(100)

Note: Numbers in parentheses ( ) are the percentage of each category.

**Table 3 foods-10-01523-t003:** Estimates of RPL model by country.

Food Values	South Korea	China	Japan
Safety	3.235 ***	4.284 ***	2.867 ***
	(0.098) a	(0.127)	(0.096)
	[1.273] ***	[1.641] ***	[0.969] ***
	(0.098) b	(0.119)	(0.092)
Taste	2.831 ***	2.441 ***	2.801 ***
	(0.102)	(0.099)	(0.100)
	[1.405] ***	[0.959] ***	[1.180] ***
	(0.094)	(0.107)	(0.087)
Nutritional Value	2.192 ***	3.044 ***	1.348 ***
	(0.088)	(0.100)	(0.081)
	[0.995] ***	[0.695] ***	[0.549] ***
	(0.084)	(0.109)	(0.104)
Origin	2.021 ***	0.087	1.826 ***
	(0.100)	(0.102)	(0.110)
	[1.525] ***	[1.399] ***	[1.737] ***
	(0.083)	(0.097)	(0.095)
Price	1.598 ***	1.291 ***	2.535 ***
	(0.096)	(0.109)	(0.121)
	[1.339] ***	[1.489] ***	[2.103] ***
	(0.083)	(0.088)	(0.125)
Naturalness	1.261 ***	2.938 ***	0.904 ***
	(0.088)	(0.110)	(0.086)
	[1.111] ***	[1.365] ***	[0.772] ***
	(0.082)	(0.096)	(0.095)
Environmental Impact	1.230 ***	1.630 ***	0.743 ***
	(0.079)	(0.095)	(0.079)
	[0.267] **	[0.720] ***	[0.156]
	(0.125)	(0.102)	(0.202)
Convenience	1.059 ***	0.816 ***	1.098 ***
	(0.083)	(0.092)	(0.085)
	[0.830] ***	[0.774] ***	[0.841] ***
	(0.094)	(0.106)	(0.096)
Fairness	0.895 ***	0.624 ***	0.362 ***
	(0.077)	(0.087)	(0.080)
	[0.604] ***	[0.698] ***	[0.792] ***
	(0.096)	(0.101)	(0.081)
Animal Welfare	0.593 ***	0.006	0.260 ***
	(0.084)	(0.100)	(0.079)
	[1.037] ***	[1.552] ***	[0.781] ***
	(0.084)	(0.100)	(0.090)
Appearance	0.460 ***	0.391 ***	0.885 ***
	(0.079)	(0.095)	(0.083)
	[0.736] ***	[1.162] ***	[0.736] ***
	(0.081)	(0.095)	(0.095)
N individuals	379	363	361
Log Likelihood Function	−8314.2	−6994.6	−7886.9

Note: Numbers in parentheses ( ) a are standard errors of mean importance of the value. Number in brackets [ ] are standard deviation. Numbers in parentheses ( ) b are standard errors of standard deviation of the value. An ** denotes significance at the 5%, and *** denotes significance at the 1% level.

**Table 4 foods-10-01523-t004:** Share of preferences of each food value by country.

Food Values	Korea	China	Japan
Safety	0.281 a	0.444	0.210
	[0.257, 0.305] b	[0.410, 0.478]	[0.186, 0.233]
Taste	0.215	0.087	0.213
	[0.190, 0.240]	[0.072, 0.102]	[0.189, 0.237]
Origin	0.126	0.014	0.131
	[0.106, 0.146]	[0.010, 0.018]	[0.110, 0.152]
Nutritional Value	0.110	0.138	0.043
	[0.094, 0.125]	[0.120, 0.157]	[0.036, 0.051]
Price	0.073	0.042	0.231
	[0.061, 0.085]	[0.033, 0.051]	[0.202, 0.260]
Naturalness	0.049	0.165	0.031
	[0.040, 0.058]	[0.142, 0.188]	[0.024, 0.037]
Convenience	0.035	0.017	0.040
	[0.028, 0.041]	[0.013, 0.021]	[0.033, 0.047]
Environmental Impact	0.030	0.035	0.021
	[0.025, 0.035]	[0.028, 0.042]	[0.017, 0.025]
Animal Welfare	0.026	0.017	0.017
	[0.021, 0.031]	[0.012, 0.023]	[0.013, 0.020]
Fairness	0.025	0.013	0.019
	[0.020, 0.029]	[0.010, 0.016]	[0.015, 0.022]
Appearance	0.017	0.016	0.030
	[0.014, 0.021]	[0.012, 0.020]	[0.025, 0.036]
Novelty	0.008	0.005	0.009
	[0.007, 0.009]	[0.004, 0.006]	[0.008, 0.01]

Note: a Numbers are mean. b Numbers in brackets [ ] are 95% confidence interval.

**Table 5 foods-10-01523-t005:** Frequency of food cold chain data by source.

	Journal	News	Blog	Total
Farm	167	386,498	152,988	539,653
	(15.9)	(42.8)	(35.8)	(40.5)
Process	265	165,255	104,718	270,238
	(25.2)	(18.3)	(24.5)	(20.3)
Package	70	61,517	32,207	93,794
	(6.7)	(6.8)	(7.5)	(7.0)
Transportation	152	45,141	23,309	68,602
	(14.4)	(5.0)	(5.4)	(5.2)
Storage	165	29,326	23,676	53,167
	(15.7)	(3.3)	(5.5)	(4.0)
Retail	69	27,420	11,794	39,283
	(6.6)	(3.0)	(2.8)	(3.0)
Customer	164	186,928	79,139	266,231
	(15.6)	(20.7)	(18.5)	(20.0)
Total	1052	902,085	427,831	1,330,968
	(100.0)	(100.0)	(100.0)	(100.0)

Note: Numbers in parentheses ( ) are the percentage of each category.

**Table 6 foods-10-01523-t006:** Supply chain node frequency and share by food value.

	Price	Safety	Taste	Nutritional Value	Origin	Naturalness
Farm (&food)	33,142	17,380	13,944	7794	5407	4960
	(12.9)	(12.9)	(17.9)	(17.6)	(17.0)	(19.5)
Process (&food)	57,896	42,120	22,847	16,253	11,397	7646
	(22.5)	(31.2)	(29.4)	(36.6)	(35.8)	(30.1)
Package (&food)	27,287	14,491	8557	5470	3842	2302
	(10.6)	(10.7)	(11.0)	(12.3)	(12.1)	(9.0)
Transportation (&food)	20,686	13,570	2841	2583	1732	2439
	(8.0)	(10.1)	(3.7)	(5.8)	(5.4)	(9.6)
Storage (&food)	17,446	7873	4740	3083	2012	2038
	(6.8)	(5.8)	(6.1)	(6.9)	(6.3)	(8.0)
Retail (&food)	16,114	3648	1900	846	611	1246
	(6.3)	(2.7)	(2.4)	(1.9)	(1.9)	(4.9)
Customer (&food)	84,611	35,855	22,940	8373	6864	4813
	(32.9)	(26.6)	(29.5)	(18.9)	(21.5)	(18.9)
Total frequency	257,182	134,937	77,769	44,402	31,865	25,444
	(100.0)	(100.0)	(100.0)	(100.0)	(100.0)	(100.0)

Note: Numbers in parentheses ( ) are the percentage of each category.

**Table 7 foods-10-01523-t007:** Food value frequency and share by supply chain node.

	Price	Safety	Taste	Nutritional Value	Origin	Naturalness	Total
Farm (&food)	33,142	17,380	13,944	7794	5407	4960	82,627
	(40.1)	(21.0)	(16.9)	(9.4)	(6.5)	(6.0)	(100.0)
Process (&food)	57,896	42,120	22,847	16,253	11,397	7646	158,159
	(36.6)	(26.6)	(14.4)	(10.3)	(7.2)	(4.8)	(100.0)
Package (&food)	27,287	14,491	8557	5470	3842	2302	61,949
	(44.0)	(23.4)	(13.8)	(8.8)	(6.2)	(3.7)	(100.0)
Transportation (&food)	20,686	13,570	2841	2583	1732	2439	43,851
	(47.2)	(30.9)	(6.5)	(5.9)	(3.9)	(5.6)	(100.0)
Storage (&food)	17,446	7873	4740	3083	2012	2038	37,192
	(46.9)	(21.2)	(12.7)	(8.3)	(5.4)	(5.5)	(100.0)
Retail (&food)	16,114	3648	1900	846	611	1246	24,365
	(66.1)	(15.0)	(7.8)	(3.5)	(2.5)	(5.1)	(100.0)
Customer (&food)	84,611	35,855	22,940	8373	6864	4813	163,456
	(51.8)	(21.9)	(14.0)	(5.1)	(4.2)	(2.9)	(100.0)

Note: Numbers in parentheses ( ) are the percentage of each category.

**Table 8 foods-10-01523-t008:** Pearson correlation coefficient between nodes of food cold chain.

	Farm	Process	Package	Transportation	Storage	Retail	Customer
Farm	1						
Process	0.92	1					
Package	0.67	0.83	1				
Transportation	0.83	0.94	0.92	1			
Storage	0.87	0.97	0.79	0.89	1		
Retail	0.66	0.68	0.63	0.71	0.67	1	
Customer	0.86	0.93	0.91	0.96	0.88	0.81	1

**Table 9 foods-10-01523-t009:** Pearson correlation coefficient between food values.

	Price	Safety	Taste	Nutritional Value	Origin	Naturalness
Price	1					
Safety	0.89	1				
Taste	0.96	0.81	1			
Nutritional Value	0.94	0.79	0.94	1		
Origin	0.93	0.95	0.88	0.86	1	
Naturalness	0.94	0.8	0.96	0.93	0.86	1

**Table 10 foods-10-01523-t010:** Correlation between food values and food supply chain.

	Farm	Process	Package	Transportation	Storage	Retail	Customer
Price	0.95	0.98	0.78	0.92	0.93	0.71	0.93
Safety	0.81	0.92	0.94	0.96	0.86	0.7	0.97
Taste	0.94	0.94	0.69	0.85	0.91	0.68	0.85
Nutritional Value	0.9	0.93	0.69	0.83	0.88	0.61	0.83
Origin	0.86	0.96	0.91	0.96	0.92	0.68	0.95
Naturalness	0.89	0.94	0.69	0.84	0.91	0.64	0.83

**Table 11 foods-10-01523-t011:** Supply-chain management strategy to ensure safety in Korea, China, and Japan.

Safety	Item	Strategy	Expected Effect
Farm	-		
Process	0.92 *	Trace records of food so that harmful substance is not added during processing	Enhance the reputation of processors for safety
Package	0.94 *	Trace records using QR codes and labels on packages of perishables	Promote track and trace
Transportation	0.96 *	Maintain proper temperature and humidity pursuant to the guideline so that perishables do not have to be discarded	Maintain safety and freshness of food
Storage	0.86 *	Maintain proper temperature and humidity for each product so that the quality of the perishable is not compromised	Maintain safety and freshness of food
Retail	-		
Customer	0.97	Trace record of perishables based on blockchain technology when consumers buy the product	Enhance consumers’ trust towards food

Note: * denotes Pearson correlation coefficients between food value and supply chain is 0.9 or higher (with round-up).

**Table 12 foods-10-01523-t012:** Supply-chain management strategy to manage food price in Japan.

	Item	Strategy	Expected Effect
Farm	0.95 *	Manage production volume and energy cost with IoT sensors	Reduce production cost
Process	0.98 *	Minimize defect rate by innovating processing techniques	Reduce production cost
Package	-	-	-
Transportation	0.92 *	Improve vehicle loading rate with shared transportation and delivery strategy	Lower logistics cost
Storage	0.93 *	Reduce inventory by building and operating large-scale integrated warehouse	Lower logistics cost
Retail	-	-	-
Customer	0.93 *	Run consumer survey on expected price	Increase purchase volume

Note: * denotes Pearson correlation coefficients between food value and supply chain is 0.9 or higher (with round-up).

**Table 13 foods-10-01523-t013:** Supply-chain management strategy to manage naturalness in China.

	Item	Strategy	Expected Effect
Farm	0.89 *	Introduce production tracking system, including soil and feed at farm and animal welfare production	Nurture quality producers
Process	0.94 *	Establish processing system that blocks alien substance and track and trace system for food additives	Increase sales of quality processors
Package	-	-	
Transportation	-		
Storage	0.91 *	Track records by attaching QR code and labelling at the package of perishables at warehouse	
Retail	-		
Customer	-		Enhance consumers’ trust towards food

Note: * denotes Pearson correlation coefficients between food value and supply chain is 0.9 or higher (with round-up).

**Table 14 foods-10-01523-t014:** Supply-chain management strategy to manage taste in Korea and Japan.

	Item	Strategy	Expected Effect
Farm	0.94 *	Buy fresh, raw produce by directly transacting with quality local producers	Secure fresh produce at the optimal price
Process	0.94 *	Develop processing techniques that maximizes texture and flavor of raw produces and extend shelf life	Maintain freshness for a certain period
Package	-	-	
Transportation	0.85 *	Maintain proper temperature and humidity and freshness during transportation considering transportation period	Keep freshness and various tastes
Storage	0.91 *	Find and maintain proper temperature and humidity that keeps products fresh and enhance flavor customized to customers	Keep freshness and age food to various tastes
Retail	-		
Customer	0.85 *	Define taste (degree of ageing, etc.) that each customer group prefers	Motivate customers to buy the product again by enhancing customers’ loyalty towards the product

Note: * denotes Pearson correlation coefficients between food value and supply chain is 0.9 or higher (with round-up).

**Table 15 foods-10-01523-t015:** Supply-chain management strategy to manage origin in Korea and Japan.

	Item	Strategy	Expected Effect
Farm	0.86 *	Manage quality at the producer and importer of imported food	Secure fresh produce at reasonable price
Process	0.96 *	Check and manage if there is no forging of certificate of origin at the stage of processing	Facilitate management of origin
Package	0.91 *	Attach and manage certificate of origin of import	Facilitate management of origin
Transportation	0.96 *	Select the mode of transportation considering the distance with the exporting country and characteristics of the product	Facilitate management of origin
Storage	0.92 *	Trace if there is no change in origin of import at the storage stage	Enhance trust towards perishables
Retail	-		
Customer	0.95 *	Provide consumers with the opportunity to choose food after seeing the values of imported production and local products	Provide wider range of choices to consumers (local vs. import)

Note: * denotes Pearson correlation coefficients between food value and supply chain is 0.9 or higher (with round-up).

**Table 16 foods-10-01523-t016:** Supply-chain management strategy to manage nutritional value in China.

	Item	Strategy	Expected Effect
Farm	0.90 *	Introduce production tracking system, including soil and feed at farm, breeding environment	Production of nutritious and healthy foods
Process	0.93 *	Monitor and track the addition of compounds to increase the content of nutrients in food, such as protein; enhance relationship between farmers and processors in industry value chain	Securing human health
Package	-	-	
Transportation	-		
Storage	0.88 *	Manage temperature, humidity, packaging to prevent food nutrition from being destroyed during storage	
Retail	-		
Customer	-		Enhance consumers’ trust towards food and securing health

Note: * denotes Pearson correlation coefficients between food value and supply chain is 0.9 or higher (with round-up).

## Data Availability

Not applicable.

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
