# Peer review of "How Do Consumers’ Food Values across Countries Lead to Changes in the Strategy of Food Supply-Chain Management?"

_foods, 2021, doi:10.3390/foods10071523_

Round 1

Reviewer 1 Report

The current version aims at answering two distinct research questions (relevance of values and connection of values with supply chain nodes). The secondo question is intriguing. I think that the current version of the paper aims at covering a two large field of investigation. My suggestion is to concentrate only on the first research question. This also beceause there is not clear illustration of why a connection there should be among values and SC nodes.

Secondly, the meaning of "price" as value has to be explained.

The work Rokeach, M. (1973). The nature of human values. Free press. should be considered.

It seems to me that there is no utility in including a table in Korean language, while the readers would benefit of an example of the choice submitted to the responded according to BIDB.

Reviewer 2 Report

The study carried out by Joo and Lee presents interesting results. However, a lot of improvements, mainly on the Discussion and Conclusions sections need to be done.

The main quantified results need to be pointed out in the abstract.

It would be interesting if the authors could provide a flowchart in section 2, with all the steps taken in carrying out the present study.

The Discussion has to be a lot of improvement. The authors should discuss the results of the present study with other studies. There is not a single reference in this section, which is unacceptable.

The study limitations need to be pointed out at the end of the Discussion.

A conclusion section is missing.

Round 2

Reviewer 2 Report

Thank you for addressing all my suggestions.